# Quantum Mechanical Study of Oxygen Ligands Protonation for the Stable States of the Laccase Active Site

**DOI:** 10.3390/ijms24032990

**Published:** 2023-02-03

**Authors:** Sergei Gavryushov, Nikolay N. Kuzmich, Konstantin M. Polyakov

**Affiliations:** 1Engelhardt Institute of Molecular Biology, Russian Academy of Sciences, Vavilova Str. 32, 119334 Moscow, Russia; 2Institute for Translational Medicine and Biotechnology, Sechenov First Moscow State Medical University, 2-4 Bolshaya Pirogovskaya Str., 119991 Moscow, Russia; 3Smorodintsev Research Institute of Influenza, WHO National Influenza Centre of Russia, 15/17 Professor Popov Str., 197376 Saint-Petersburg, Russia

**Keywords:** laccase, enzymatic oxygen reduction, reaction mechanism, DFT calculations, oxygen ligands protonation

## Abstract

Laccases are enzymes catalyzing the oxidation of a wide range of organic and inorganic substrates accompanied by molecular oxygen reduction to water. Recently, oxygen reduction by laccases has been studied by single-crystal serial X-ray crystallography with increasing absorption doses at subatomic resolution. There were two determined structures corresponding to the reduced and oxidized stable states of the laccase active site. However, the protonation of the oxygen ligands involved cannot be determined even at subatomic resolution. In the present work, the protonation of oxygen ligands in the active site of laccase for the two stable states determined in the X-ray study was explored using quantum mechanical and continuum-electrostatics calculations. This is important for understanding the reaction of the oxygen reduction mechanism in laccases. The high precision of X-ray data at subatomic resolutions allowed us to optimize the quantum mechanical calculations.

## 1. Introduction

Laccases are multi-copper oxidases that catalyze the oxidation of different substrates, such as *p*-diphenols, while simultaneously reducing molecular oxygen to water. Laccases have been found in bacteria, fungi, plants, and insects. Laccases were studied by spectroscopic methods [1], and then their structures were determined by X-ray crystallography [2] (review of results up to 2015).

Laccases are globular proteins comprised of two or three domains of cupredoxin fold [2]. Their active site includes four copper ions arranged in two separated centers: the T1 copper site and the trinuclear copper cluster (TNC). Three types of copper ions [3] in laccases were well-established in earlier studies [1,4]. The TNC contains a triangle of closely located copper ions, including a pair of T3 copper ions coordinated by three histidine residues each, and one T2 copper ion coordinated by two histidine residues. Electrons are accepted from substrate at the T1 copper site. Via an electron-transfer chain electrons are transferred to the TNC where the molecular oxygen reduction reaction O_2_ + 4e^−^ + 4H^+^ → 2H_2_O occurs [1,5]. The TNC is connected to the surrounding solvent via the T3 and T2 channels clearly observed in structures as chains of water molecules [6] (Figure 1). Molecular oxygen gets to the TNC through the T3 channel.

The molecular oxygen reduction at the TNC was considered in works by Solomon and coworkers [4,5] based on spectroscopy and early X-ray studies of ascorbate oxidase [7]. It includes a couple of two-electron reduction stages. Among the TNC states at the oxygen reduction, the following ones can be distinguished. In the fully reduced state of the TNC, all three copper ions are reduced to charge +1e and two oxygen ligands are involved (one between two T3 copper ions outside TNC, and the other one near the T2 copper ion in channel T2). The peroxy intermediate (PI) state includes deprotonated hydrogen peroxide located inside the copper ion triangle. The T2 and one of the T3 copper ions are oxidized to Cu(II), and there is an oxygen ligand near the T2 copper ion. In the native intermediate (NI) state, all three copper ions are oxidized to charge +2e, and there are three oxygen ligands involved. One is the oxygen ligand near ion T2 in the channel, one is in the center of the copper ions triangle, and one is between two copper ions T3. The last two appear from peroxide when the O–O bond is broken. Each of the T3 copper ions is five-coordinated by three nitrogen atoms of histidine residues and by two oxygen ligands between the T3 copper ions in the TNC.

The fully reduced and NI states are clearly observed from X-ray data [6], whereas the PI state has been deduced entirely from spectroscopic data. The features of this oxygen reduction scheme are the asymmetric location of the two oxygen ligands in respect to copper ions T3 in the PI state, and the involvement of proton transfers between an aspartic acid residue in the T2 channel and the oxygen ligand near the T2 copper ion. This leads to the appearance of a hydroxyl ion instead of a water molecule as the T2 copper coordinating oxygen ligand in channel T2 [8]. The fully reduced state is stable, whereas the NI decays, but this process is very slow [9].

Based on the above-mentioned scheme of molecular oxygen reduction in laccases [5], studies were conducted using quantum mechanical (QM) and molecular mechanical methods [10,11,12,13]. Those QM calculations based on the density functional theory (DFT) agreed with the suggested reaction mechanism and demonstrated a strong dependence of obtained energies on the protonation state of the oxygen ligands coordinating the TNC copper ions [12,13]. However, the oxygen ligand transfer from the center of the trinuclear cluster to the copper coordinating position in the T2 channel [12] is questionable. According to high-resolution X-ray data structures of fungal laccase [6], surrounding residues involved in the coordination of the TNC copper ions leave no space for such a move because of steric restrictions in the trinuclear center.

Recently, there have been reported serial structures of laccase with different degrees of TNC copper ions reduction solved at high resolution for sets of X-ray data collected from one crystal [6,14]. These studies, as well as other reported results for serial crystallography [15,16], revealed a picture that contradicted, to some extent, the scheme of oxygen reduction suggested earlier. Due to the high quality of subatomic resolution data for sets of structures with different degrees of oxidation, both oxidized and reduced states of the TNC were resolved and clearly interpreted. As a result, it was possible to determine the coordination of the TNC copper ions in the oxidized (a) and reduced (b) states of the TNC (Figure 1). There were two detected positions of each copper T3 ion corresponding to its Cu(I) or Cu(II) state. In state Cu(I), the copper ion is three-coordinated by three histidine residues. In the oxidized state, Cu(II) is five-coordinated by those histidine residues and two oxygen ligands. In the reduced state, the distance between the T3 ions is about 5.2 Å, and it is about 3.3 Å in the oxidized state. For the oxygen ligand coordinating the T2 copper ion in channel T2, there was detected switching between two ligand positions corresponding to the two states, which clearly indicates a change of the T2 copper ion coordination at its oxidation. In the reduced TNC state, Cu(I) ion T2 is linearly coordinated to nitrogen atoms of two surrounding histidine residues. The neighboring oxygen ligand in the T2 channel interacts with copper ion T2 via electrostatic forces. When an oxygen ligand is found at the center of the trinuclear cluster, the T2 Cu(II) ion has square-planar four-coordination with the addition of this oxygen ligand and the oxygen ligand in the T2 channel. As a result, the distance between the T2 copper ion and the oxygen ligand in the T2 channel diminishes.

Based on those serial X-ray data, some corrections to the oxygen reduction scheme were discussed [6]. In this reaction model, an oxygen molecule, being an induced dipole attracted by the trinuclear cluster of ions Cu(I), penetrates between two copper ions T3, oxidizing both of them. The appearing PI state is symmetrical in respect to the five-coordinated T3 ions. It will wait for the next coming electron that reduces one of the T3 copper ions. The following cleavage of the peroxide O–O bond is caused by the oxidation of this copper ion and copper ion T2 by the two oxygen atoms and the change in coordinating the T3 and T2 copper ions. The final structure corresponds to the NI state of the TNC. The next reductions of TNC copper ions due to the electron supply release two oxygen ligands from the copper ion cluster, restoring the reduced state of the TNC. According to this scheme, the change of T2 copper ion coordination is accompanied by the cleavage of the peroxide O–O bond. In this scheme, all of the stable TNC states are quasi-symmetrical structures in respect to the pair of copper ions T3. Copper ion T2 is oxidized last, and there is no need in appearance of a hydroxyl ion in the T2 channel near the T2 copper ion.

The oxidized (NI) and reduced states of the TNC were well-interpreted from the serial X-ray data. Positions of the TNC atoms and ligands in the oxidized state of the TNC (the native intermediate) are shown in Figure 1a. The oxygen ligand in the center of the copper triangle is denoted as W1, the oxygen ligand in channel T2 as W2, and the oxygen ligand in channel T3 as W3. In the reduced state, ion Cu(I) T2 is linearly coordinated by two nitrogen atoms of the histidine residues, and oxygen ligand W2 is connected to the ion Cu(I) by only ion-dipole electrostatic interaction. In the fully oxidized state, position W2 is closer to the ion due to involvement of the oxygen ligand W2 into the T2 Cu(II) coordination. Positions Cu(I) T2 and Cu(II) T2 almost coincide.

In the previous quantum mechanical calculations for TNC models [13], there was shown a strong dependence of the QM-calculated energies of states at cleavage of the O–O bond upon protonation of the oxygen ion in the center of the trinuclear cluster. It indicates an importance of true protonation states of the oxygen ligands involved. Protons cannot be resolved from a solution of X-ray data structures where the protonation of the oxygen ligands remains undetermined even at atomic resolution.

In the present work, DFT calculations have been applied for a study of the protonation of the TNC oxygen ligands (Figure 1) for the observed stable oxidized (NI) and fully reduced states of TNC based on serial one-crystal X-ray data at subatomic resolution [6,14]. The QM calculations have also been supplemented by continuum-electrostatics calculations of energies of the proton migration in the outer parts of the laccase’s channels. Based on the determined protonation of the TNC oxygen ligands, the scheme of the oxygen reduction mechanism has been improved.

## 2. Results

### 2.1. Results of DFT Calculations

#### 2.1.1. Protonation of the W1 Oxygen Ligand in the Center of TNC

The model for DFT calculations to explore protonation of oxygen ligand W1 in the NI state of TNC is shown in Figure 2. The proton transfer from ligand W3 (H_2_O) to the central oxygen ligand W1 (O^2−^) is considered.

The model includes atoms of eight histidine residues. Residues His 67, His 454, His 110, His 402, His 112, and His 452 coordinate the pair of T3 copper ions in the oxidized state. Residues His 65 and His 400 coordinate the T2 copper ion. For DFT calculations, the polypeptide chain was truncated, and each beta carbon of the residues was replaced by a methyl cap. The model includes three TNC copper ions in the oxidized state Cu(II) and four oxygen ligands from the X-ray data structure corresponding to the oxidized state of the TNC. Three of them are ligands W1, W2, and W3 (Figure 1a), and the fourth one is the oxygen ligand near W3 in the T3 channel. In all calculations, the oxygen atoms of ligands W1 and W2 were constrained, whereas ligand W3 was subjected to geometry optimization. Due to steric restrictions inside the trinuclear cluster, two positions of hydrogen of the central OH^−^ ion are only possible: “above” and “below” the plane of copper ions triangle, almost perpendicular to the plane. Both of them have been studied in following geometry optimization and DFT calculations. In the following text, they will be called positions “up” and “down.” Further in this section, we denote the state of the system according to the protonation state of oxygen ligands W1 and W3 as (W1, W3) (for example, (O^2−^,H_2_O)).

The study included several DFT calculations. Two were done with the central oxygen ligand (as O^2−^ and OH^−^ ion) released to determine its calculated positions. A particular QM study was pointed to determine the transition state at the proton transfer from H_2_O at position W3 to O^2−^ at position W1 in order to evaluate the potential barrier of this reaction. Finally, energies of the three states ((O^2−^,H_2_O), (OH^−^up,OH^−^), (OH^−^down,OH^−^)), and an energy barrier at the proton transition from W3 to W1 were calculated.

The QM calculations revealed that the W1 O^2−^ ion of the oxidized TNC form should get a proton from oxygen ligand W3 if the latter one is a water molecule. We considered: (i) the coordinate shifts of both protonated (OH^−^) and deprotonated (O^2−^) ions in respect to the T2 copper atom when constraint was removed for W1 oxygen specimen; (ii) the energy of the proton transfer from the W3 water molecule to the W1 O^2−^ oxygen ligand. The results of the QM calculations are as follows.

(i) The DFT-determined position of ligand W1 was found close to its position in the X-ray data structure. In the case of released protonated ligand W1, the oxygen atom shift, relative to the T2 copper ion position, accounted for 0.03 Å compared to the X-ray structure. Such a shift consisted about 0.12 Å for deprotonated oxygen O^2−^.

(ii) The results of the QM energy calculations for different states of the system are collected in Table 1. Using the QST3 algorithm [17], the transition state of proton transfer from water molecule W3 to fully deprotonated ion of oxygen O^2−^ at position W1: H_2_O + O^2−^ → 2OH^−^ was studied. The potential barrier is high enough (~18 kcal/mol), but the final state (OH^−^,OH^−^) has a noticeable lower energy than the initial state (decreased by ~−20 kcal/mol), and there is a relatively small difference between the two states of the hydroxyl ion (hydrogen atom “up” and “down”) of about 6 kcal/mol.

It should be noted that we used H_2_O as ligand W2 (Figure 2), but in a case of OH^−^ instead of H_2_O, this energy difference in a favor of a hydroxyl ion at W1 would only be higher, as the proton is additionally attracted by the negatively charged hydroxyl ion at W2.

Thus, according to the DFT calculations, the W1 oxygen ligand will be protonated for this model.

#### 2.1.2. Protonation of W3 Oxygen Ligand in the Oxidized State of TNC

The molecular model for DFT calculations is shown in Figure 3. The transfer of a proton from the neighboring oxygen ligand in the T3 channel as a hydroxonium cation to W3 OH^−^ was considered. The protonation of the oxygen ligand at position W3 is estimated via the difference of DFT-calculated energies of these two states. The input structure in the calculation setup comprised the T3 channel, including the acetate fragment of Asp 456, the phenol group of Tyr 117 as a hydrogen-bond donor, and the extended part of His 454 residues. The O-H bond lengths in hydroxonium cation were kept fixed throughout the optimization and were set to 1.023 Å [18]. The model was constructed so all molecular fragments involved can form hydrogen bonds with water molecules and hydronium ions chosen in channel T3 or interact with them via long-range electrostatics.

The DFT calculation results suggest that the system energy grows at the proton transfer from hydroxonium cation located at a position of the next towards T3 channel entrance oxygen atom to the hydroxyl ion at position W3: the energy difference consists of +6.28 kcal/mol. Thus, according to this calculation, oxygen ligand W3 should be treated as OH^−^.

#### 2.1.3. Protonation of Oxygen Ligand W3 in the Reduced State of TNC

The protonation of W3 in the reduced state of the TNC (Figure 1b) was studied via DFT-calculated energies for a system similar considered for oxidized TNC. The only difference is, in the absence of the W1 oxygen ligand, all three copper ions are reduced to Cu(I), copper ions T3 have different positions, and both W2 and W3 are retained via charge-dipole electrostatic interactions (Figure 4). In this case we consider two proton transfers in the channel, i.e., energies of three systems are calculated.

The DFT-calculated results unambiguously suggest in favor of a water molecule as ligand W3. The energy differences are given in Table 2 where we describe the state as (W3+2,W3+1,W3), with W3+1 and W3+2 denoting the first and second oxygen ligand from the W3 oxygen ligand in the channel, respectively (Figure 4).

#### 2.1.4. Protonation of the W2 Oxygen Ligand in the Reduced State of TNC

In this section, we describe the results of DFT calculations for the reduced state of TNC (Figure 1b). To determine the protonation of oxygen ligand W2, energies of states with the W2 oxygen ligand as H_2_O and OH^−^ are compared. The TNC environment chosen as a model of DFT calculations is shown in Figure 5. Energies of three states of the system are calculated. In this section, we will denote each state as (W2+2,W2+1,W2) with W2+1 and W2+2 denoting first and second oxygen ligand from oxygen ligand W2 in the channel, respectively. Then, the three states considered will be (H_3_O^+^,H_2_O,OH^−^), (H_2_O,H_3_O^+^,OH^−^), and (H_2_O,H_2_O,H_2_O).

The model includes two protein chains surrounding channel T2. Uncharged amino acid side chains were replaced by methyl groups. Residues and backbone atoms involved into forming a net of hydrogen bonds with water molecules in the channel are described in more detail. The positions of water molecules were taken from the X-ray data structure. Five water molecules in the T2 channel are included in the model: three ones on the way to the exit from the channel (including ligand W2) and two ones linking the W2 ligand and Asp 78 connected to His 402 by a non-covalent bond (His 402, in return, coordinates one of the T3 copper ions). The last aspartic acid was assumed to play an important role as a proton acceptor in the scheme of oxygen reduction in the TNC discussed earlier [5]. In our QM study, we considered just a transfer of a proton in the T2 channel from outside. We compared the energy of two states: the second water molecule from the W2 ligand (OH^−^) is a hydroxonium cation (H_3_O^+^) versus two H_2_O molecules as these two ligands. In other words, we considered a proton transfer along the T2 channel to the hydroxyl ion as ligand W2 in the reduced state of the TNC. Now, the starting position of W2(I) is taken from coordinates interpreted for the reduced TNC (pdb entry 6RGP, Figure 1b) and ligand W3 is treated as a water molecule (Figure 5). One more water molecule next to W3 is also included in the model.

The results of calculations TPSSh/6-31G** (Section 4 Materials and Methods) also suggest in favor of a water molecule at position W2(I) for the reduced state of TNC. Indeed, a proton transfer from the third oxygen ligand in the T2 channel to hydroxyl ion at position W2(I) is energetically quite favorable (36.2 kcal/mol). Results of energy DFT calculations are given in Table 3. It should be noted that the replacement of a water molecule as ligand W3 by a hydroxyl ion only increases this difference in favor of H_2_O as the W2 ligand since a hydroxyl ion attracts the transferred proton.

There were performed geometry optimizations without any spatial constraints for oxygen ligand W2 for both states of its protonation (OH^−^ and H_2_O). As follows from those optimizations, the shift of the oxygen atom from its X-ray-determined position towards Cu(I) ion T2 is negligible in case of H_2_O (0.02 Å), whereas the hydroxyl ion appeared notably closer to the T2 ion (shift of 0.32 Å), which is beyond statistical uncertainty of the structure solution from X-ray data. Thus, these results of optimizations are in accordance with the interpretation of oxygen ligand W2 as H_2_O.

#### 2.1.5. Protonation of the W2 Oxygen in the Oxidized State of TNC

The model to estimate the protonation of the W2 oxygen ligand in the oxidized state of the TNC (Figure 1a) is similar to the one considered in the previous section except for the copper ions charge (+2e). The two models are also different in respect of coordinates of TNC copper ions and oxygen ligands W1, W2, and W3 taken from the TNC oxidized state interpreted from X-ray data (pdb entry 6RGP). The model is shown in Figure 6. Similar to Section 2.1.4, there were calculated energies of the system at a consequent transfer of a proton from the second water molecule in the T2 channel to a hydroxyl ion at position W2(II). As before, we describe the state as (W2+2,W2+1,W2), where W2+2 is the second oxygen ligand from oxygen ligand W2 and W2+1 is the oxygen ligand next to the W2 oxygen ligand in the T2 channel.

The results of the DFT calculations for the proton transfer shown in Figure 6 are given in Table 4.

They suggest in favor of a hydroxyl ion as the W2 oxygen ligand when the TNC is oxidized. The energy difference between states (H_2_O,H_2_O,H_2_O) and (H_3_O^+^,H_2_O,OH^−^) is +3.8 kcal/mol. However, this value is relatively low and the final conclusion about the protonation of this ligand can be drawn only after a consideration of the mean field created by the excluded external part of the protein and the energetic cost of the proton transfer in the outer part of the T2 channel (see Section 3 Discussion).

#### 2.1.6. Protonation of the W3 Oxygen in the Oxidized State of TNC after the Cleavage of the Oxygen Peroxide Bond

According to the suggested reaction mechanism [6], the protonation of the oxygen ligands in the NI state requires three proton transfers through channel T3. One transfer is considered in Section 2.1.2 (ligands W3 and W3+1 become water molecules when a hydroxyl ion is at position W1). One needs to consider two more proton transfers through channel T3. It is necessary to consider the protonation of oxygen ligands in channel T3 after the cleavage of the bond between oxygen atoms in the hydrogen peroxide molecule. According to the reaction scheme, this cleavage is caused by a coming electron, when one of the T3 ions Cu(II) gets reduced. It pushes the deprotonated peroxide O^−^–O^−^ inside the TNC, this T3 copper ion and copper ion T2 get oxidized along with switching the coordination of the copper ion T2 from linear to square planar. As a result, the bond between oxygen atoms of peroxide is broken, and there are two ions O^2−^ at positions W1 and W3. Next, a proton from neighboring water at position W3+1 moves to ion O^2−^ at position W3 and two hydroxyl ions appear at positions W3 and W3+1. It results in a state for the triangle of fully oxidized copper ions Cu(II) where there are ion O^2−^ at position W1 and two hydroxyl ions at positions W3 and W3+1. As follows from PB calculations (see below), the hydroxyl ion at position W3+1 can be protonated easily. Thus, one needs to consider the second transfer of a proton in channel T3 to get a water molecule instead of a hydroxyl ion at position W3 (i.e., the initial state for the proton transfer between oxygen ligands W3 and W1 described in Section 2.1.1).

In this section, the DFT calculations were applied to study such a proton transfer in channel T3. The molecular model is shown in Figure 7. The model is quite similar to the one described in Section 2.1.2. Like in Section 2.1.3, we describe the state as (W3+2,W3+1,W3). The results are summarized in Table 5. As follows from DFT calculations, the proton transfer from position W3+2 to a hydroxyl ion at position W3 is energetically favorable (−10 kcal/mol).

### 2.2. Results of Calculations for the Continuum-Electrostatics Model

The QM calculations considered the TNC’s adjacent environment in the gas phase. Such an approach is possible due to the fact that the TNC is immersed into the hydrophobic core of the laccase macromolecule with low permittivity. However, an impact of the protein’s distant atoms on the proton transfer should be taken into account. It includes both the QM energy differences considered above and the energy of proton transfer from bulk solution to the position of a hydroxonium ion from which the proton transfer was studied by means of QM calculations. Despite the wide application of the DFT with QM/MM methodology [19] to enzymatic reactions such as e.g., hydrolysis [20], or the application of DFT to redox processes [21,22], evaluations of these contributions are not an easy task [23]. If one uses, e.g., the continuum-electrostatics approach, such as the PB approximation, it would be necessary to calculate the changes of the mean electrostatic potential between positions of the transferred proton created by the charges of the outer part of the protein macromolecule. However, the protein interior is, to some extent, a polarizable medium with an unknown spatial distribution of its mean dielectric constant. The last value is also unknown. It is estimated between 2 and 10 and usually assumed to be about 4 at its homogeneous distribution in the volume occupied by protein atoms. Since PB results are sensitive to this unknown parameter, we have eliminated unreliable evaluations of the electrostatic energy changes due to surroundings for the proton transfers studied via the QM calculations (Section 2.1). Anyhow, it should be noted that this impact could not be negligible comparatively to the QM-calculated energy differences. For instance, as follows from PB calculations of the mean electrostatic potential inside an empty cavern in the laccase macromolecule due to the removed QM model atoms, the electric field of the protein outer part and electrolyte ions can change the QM-calculated energy difference by several kcal/mol despite short (a few Ångstrom units) distances of the proton transfers.

To study the TNC oxygen ligands protonation, PB calculations have been applied to evaluate a more significant contribution due to the change of energy during a proton transfer from a far-off position outside the protein macromolecule (infinity) to the most remote positions of hydroxonium cations in laccase channels, which were used in the QM calculations. In other words, there were calculated the electrostatic energy changes at the proton transfers from infinity to positions W2+2 and W3+2 of the crystallization water in the channels T2 and T3, respectively (Section 2.1.3, Section 2.1.4, Section 2.1.5 and Section 2.1.6). The sum of the energy change at such a transfer and the QM-calculated energy difference at the final part of the proton transfer to oxygen ligands W2 and W3 gives the total energy change at proton migration from bulk solution to these ligands. This total energy change finally allows us to evaluate the p*K*_a_ of ligand’s dissociation (see section Discussion).

Each model chosen for a continuum electrostatics study was explored by several PB calculations varied in respect to parameters that were not known precisely and could only be estimated within some limits. They were the partial charges of the protein (only charged groups or also partial charges from the Amber force field), the dielectric constant of the protein interior (2–10, the mostly plausible value was 4), and the radius of the hydration of the cation describing ion H_3_O^+^ (1.8–2.3 Å). PB calculation results were sensitive enough to the latter two parameters due to the Born energy of hydration. Deeply inside the narrow channels of laccase, the PB calculations get quite inaccurate due to the roughness of the model. Therefore, such electrostatic energy differences as, e.g., between W2+2 and W2+1 positions of the hydroxonium ion, may not be regarded as realistic ones. All internal histidine residues were treated as neutral ones, whereas solvent-accessible histidine residues bore nitrogen charges +0.5e except for His 458 coordinating the T1 copper ion. In addition, their charge was chosen as +1e for comparison. Such a change of their charge had a weak effect on the calculated energy differences.

The PB calculations are described in Appendix A. The data are given in Appendix A, where results are shown as diapasons of values of the energy differences depending on the varied unknown PB calculation parameters described above. The most sensitive parameter was the dielectric constant of the protein interior varied from 2 to 10. Here, PB results are briefly summarized for the dielectric constant of protein equal to 4.

For the reduced TNC state, the change of the mean electrostatic energy at a transfer of charge +e from bulk solution to position W2+2 is low and rather negative (no less than −2 kcal/mol). It is also negative in channel T3: a transfer from outside to position W3+2 decreases the energy by several kcal/mol (up to 6 kcal/mol, Appendix A).

For the TNC in its oxidized state, a transfer of charge +e in channel T2 from bulk solution to position W2+2 increases electrostatic energy by several kcal/mol. In the case of a hydroxyl ion at position W3, this increase does not exceed 6 kcal/mol (Appendix A). For channel T3, we should consider three transfers of a proton from outside to position W3+2 in the case of the oxidized TNC copper ions Cu(II). The first one is the transfer just after the cleavage of the peroxide bond (see Section 2.1.6). Then there are ion O^2−^ at position W1 and two hydroxyl ions at positions W3 and W3+1. According to PB calculations, the transfer of charge +e from infinity to position W3+2 is energetically quite favorable. The resulting drop of electrostatic energy exceeds 10 kcal/mol (Appendix A). The second transfer precedes the initial state in the QM study described in Section 2.1.6. There are ion O^2−^ at position W1, a hydroxyl ion at position W3, and a water molecule at position W3+1 (Figure 7). Then, the transfer of charge +e from outside to position W3+2 increases the electrostatic energy by several kcal/mol (up to 9 kcal/mol, Appendix A). The third proton transfer in channel T3 from bulk solution to a water molecule at position W3+2 corresponds to the model considered in Section 2.1.2 (Figure 3). It is assumed that the oxygen ligand W1 has become a hydroxyl ion, and there is the second hydroxyl ion at position W3. Irrespective to the oxygen ligand at position W2 (hydroxyl or water), the change of electrostatic energy at such a transfer is positive and high enough (up to 18 kcal/mol, Appendix A).

## 3. Discussion

The present work is a development of the study of molecular oxygen reduction in laccases [6,14]. The previous exploration was entirely based on X-ray mono-crystal serial data with an increasing absorbed radiation dose at subatomic resolution. Those studies allowed the process of molecular oxygen reduction in TNC to be traced, and the TNC structures in their two stable states were determined (NI and reduced states, Figure 1). In the present work, the protonation of the laccase’s oxygen ligands in these two stable states is estimated via DFT calculations.

For QM calculations, the coordinates of non-hydrogen atoms were taken from X-ray data solved at subatomic resolution. Those X-ray data allowed us to assign the degree of oxidation to each copper ion position. It reduced QM optimization due to the fixation of copper ions and restrained the number of models studied compared to the previous research [10].

The present QM and PB calculations together allow us to draw some conclusions about the protonation of the oxygen ligands involved into the oxygen reduction cycle in laccases. The p*K*_a_ of dissociation of ligand will be p*K*_a_ = −log(e)Δ*G*/*RT,* where Δ*G* (in kJ/mol) is a sum of the QM and estimated electrostatic energy changes at the proton transfer from infinity to ligand. According to the Henderson-Hasselbalch relation, for the dissociation of ligand in the active site p*K*_a_ = pH − log([OH^−^]/[H_2_O]), where [OH^−^] and [H_2_O] are concentrations of the bound ligand in its deprotonated and protonated states, respectively. Then the protonated state occupancy will be *η* = [H_2_O]/([H_2_O] + [OH^−^]) = 1/(1 + 10^pH-p*K*a^). Using the definition of pH = −log(*N*_p_^0^/*N*_w_^0^) and the relation between p*K*_a_ and Δ*G*, the expression of *η* can be rewritten as *η* = *α*/(1 + *α*) where *α* = (*N*_p_^0^/*N*_w_^0^)exp(−Δ*G*/*RT*). Here, *N*_p_^0^ is the hydroxonium ion number density in the bulk solution, *N*_w_^0^ is the water molecule number density in the bulk solution, and Δ*G* is the free energy difference for the proton transfer from bulk to ligand. Assuming that laccase works in the acidic environment (pH = 5), the ratio *N*_p_^0^/*N*_w_^0^ is 10^−5^. One can rewrite the expression of *α* as exp(−(Δ*G*_0_ + Δ*G*)/*RT*), where Δ*G*_0_ = −*RT* ln(*N*_p_^0^/*N*_w_^0^). This value is +6.8 kcal/mol at temperature 298K and pH = 5. In other words, for the protonated state occupancy higher than 0.5, the energy of proton transfer from the bulk to this site must be lower than −6.8 kcal/mol at pH = 5 and even less at higher pH.

For the reduced state of TNC, both W2 and W3 oxygen ligands should be water molecules. Indeed, as follows from the QM and PB calculations, the rest of the transfer of a proton from position W3+2 to the hydroxyl ion at position W3 leads to the energy decrease of 31 kcal/mol (Table 2), whereas a proton transfer from the bulk solution to position W3+2 rather decreases the system energy more (the maximal estimate of its possible increase does not exceed 4 kcal/mol) (PB calculations, Appendix A). Thus, the calculated total decrease of the energy at protonation of a hydroxyl ion at position W3 in the reduced state of TNC exceeds 27 kcal/mol. This value cannot be compensated neither by the external field neglected in the QM calculations (a few kcal/mol) nor by the Boltzmann factor for hydroxonium ions in the bulk (6.8 kcal/mol). A similar picture is observed in channel T2: the QM calculations give the energy decrease of 36 kcal/mol (Table 3) and the preceding proton transfer from the bulk to position W2+2 (PB calculations) can diminish this value by no more than 4 kcal/mol (Appendix A). In other words, in the case of the reduced state of TNC, the entire proton migration from the bulk solution to the hydroxyl ion at position W2 in channel T2 is energetically quite favorable as well.

These results are in accordance with expectations. For oxygen ligand W3 as a hydroxyl ion, the induced polarization of an oxygen molecule would hardly be competitive with doubled energy of ionic pair OH^−^–Cu^+^ to approach the pair of copper ions T3. On the other hand, the competition between an induced dipole of molecular oxygen and a permanent dipole of a water molecule looks plausible enough.

For the oxidized state of TNC, the picture is more complicated. The oxygen ligand W3 in the NI state should be a hydroxyl ion. The proton transfer from the hydroxonium at position W3+1 to the hydroxyl ion at W3 increases energy by 6 kcal/mol as follows from the QM calculations (Section 2.1.2, Figure 3). The migration of a positive charge from bulk to position W3+2 in the T3 channel significantly increases this energy irrespective of oxygen ligand W2. Taking into account the Boltzmann factor for hydroxonium ions, one has to conclude that a hydroxyl ion at W3 cannot be protonated in the oxidized state of the TNC.

After the cleavage of the peroxide bond, a proton migration from the bulk to water at position W3+2 is quite favorable, according to the PB calculations (Section 4). Finally, it leads to the further jump of a proton from hydroxonium ion at W2+2 to hydroxyl ion at W2+1. Thus, we conclude that there can easily be achieved a state of oxidized TNC when there are ion O^2−^ at position W1, OH^−^ at position W3, and water molecules as other oxygen ligands in channel T3. 

The next step of oxygen ligands’ protonation in the oxidized state of the TNC is the second migration of a proton in channel T3 to get a water molecule instead a hydroxyl ion at position W3 when ion O^2−^ remains at position W1 (this is the starting configuration of the QM calculations described in Section 2.1.1). I.e., there are ion O^2−^ at position W1, ion OH^−^ at position W3, and molecules H_2_O at positions W3+1 and W3+2. The PB calculations indicate that a proton migration to position W3+2 meets a potential barrier in the channel despite the negatively charged aspartic acid Asp 456 on the way of the transfer. Even when we choose a hydroxyl ion at position W2 instead of the water molecule, the migration of a proton to position W3+2 can meet a barrier estimated from 1 to 9 kcal/mol (a more precise value is beyond the accuracy of our model). When we assume a more polarizable protein interior, this barrier is much lower (up to 2 kcal/mol, Appendix A). On the other hand, the rest of the proton transfer from position W3+2 to hydroxyl ion at position W3 decreases energy by 10 kcal/mol, according to the QM calculations (Section 2.1.6, Table 5). Thus, one may admit that there could appear the water molecule at position W3 needed for the proton transfer to the central oxygen ligand (Figure 2), but it seems questionable at some possible values of the unknown model parameters.

Finally, we will consider proton transfers in channel T2 when the TNC is in its oxidized state. The results of the PB calculations for the proton transfer in channel T2 were obtained for a model of the oxidized TNC with a hydroxyl ion at the centre of the copper ions triangle. As follows from these data, the proton transfer to position W2+2 in the oxidized state of TNC increases the energy of the system by several kcal/mol. Again, the accuracy of the model is low, and the true value cannot be calculated. As follows from the QM calculations, a further transfer of a proton from a hydroxonium ion at position W2+2 to a hydroxyl ion at position W2 also slightly increases the energy (about 4 kcal/mol, Table 4). Although the sum gives a growth of energy, the definite conclusion cannot be drawn. First, the lower limit of this sum is about 6 kcal/mol (Table 4 and the second line of Appendix A). Second, at such low energies, the mean field of the protein environment (neglected in the QM calculations) could affect the result by values of comparable magnitude, decreasing the proton transfer energy up to negative values. One could only say that the water molecule at position W2 can lose its proton in the oxidized state of the TNC. In such a case, this proton will migrate in the channel T2 towards exit from the channel. In addition, we have only considered a case of hydroxyl ion at position W1. If ion O^2−^ is present there, such a loss of a proton looks much less probable due to additional attraction of the proton by this ion.

Thus, a determination of the degree of protonation of the oxygen ligand at position W2 remains questionable for the oxidized state of TNC. Experimental data rather suggest in favor of a water molecule there. It follows from the fact that halogen ions are bound at this position [14], and they inhibit the molecular oxygen reduction. The plausible explanation is that their negative charge impedes the reduction of the T2 copper ion, whereas a water molecule does not [14].

## 4. Materials and Methods

All initial coordinates of non-hydrogen atoms were taken from the 6RGP record of the Protein Data Bank [14]. The 6RGP structure was solved at a subatomic resolution (0.97 Å) and presents clearly distinguishable oxidized and reduced enzyme forms (Figure 1). To model the proton transfers along the distal parts of the water wires in both channels, the QM calculations included the TNC’s adjacent environment in a gas phase. Such an approach is possible due to the fact that the TNC is immersed into the hydrophobic core of the laccase macromolecule. In all models, there were included three TNC copper ions along with side chains of eight histidine residues and oxygen ligands coordinating them.

The chosen molecular model around the active site was a compromise between delivering a minimal adequate chemical environment and computational cost. The input structures were prepared in the following way. Initially, the 6RGP structure was imported into Maestro 11.8 graphical interface within Schrodinger 2018-4 software [24]. For oxidized and reduced laccase forms, the corresponding copper and oxygen atom coordinates were chosen from X-ray data according to the analysis performed previously [14]. Then, all hydrogens of the model were added, and bond orders were assigned employing the Protein Preparation Wizard and manually edited as necessary for the model chosen. In most cases, histidine residues were truncated to their side chains with beta carbons substituted by methyl groups. When proton transfers in the T2 or T3 channels were considered, the water-coordinating functional groups (C=O, NH, COO-) were also preserved, as well as other oxygen ligands. The side chains of amino acids, other than histidine, were removed and backbone fragments were capped (N-methylated C-termini and acetylated N-termini). Aspartatic side chains were truncated to acetates. To fill up the free valences where the backbone and side chain fragments were cut off, the hydrogens were added.

All the DFT calculations were performed with Gaussian 16 Revision B 01 [25] software using TPSSh density functional (meta-hybrid density functional [26,27]). It performs well in describing reactions of transition metal systems [28,29]. The basis sets were composite: the Pople’s split-valence 3-21G* and 6-31G** for *p*-elements and effective core potentials (ECP) for copper atoms [30,31]. The pseudopotentials and basis sets themselves were taken for 10 core electrons, multi-electron approximation, and completely relativistic (ECP10MDF) [25,32].

The positions of copper ions and most of the non-hydrogen atoms of residues were taken from the X-ray data structure and kept frozen, whereas all hydrogen atoms, oxygen ligands chosen for a particular model, and carbon atoms of capping methyl groups were free. The initial geometry optimization was carried out using smaller 3-21G* basis sets for *s*- and *p*-elements. There were calculated energy second derivatives for a final optimization that was done with a larger 6-31G** basis set. Geometry optimizations followed by frequency calculations to confirm the correctness of stationary points were found, as well as thermochemistry calculations, including zero-point energy.

Oxygen ligand protonation was evaluated via a comparison of the energies of a system with different protonation of oxygen ligands. The energy barrier of a proton transfer from the W3 to W1 oxygen ligands was also estimated. The potential barrier at the proton transfer was studied by means of determining the transition state using the quadratic synchronous transit 3 (QST3) algorithm [17]. The transition state finding was verified by the intrinsic reaction coordinate (IRC) approach [33].

The continuum-electrostatics calculations were applied to estimate the electrostatic energies of the proton transfer from infinity (~17 Å apart from the protein surface) to oxygen ligands surrounding the copper ions T2 and T3. The calculations were based on solutions of the Poisson-Boltzmann (PB) equation for an atom model (PDB entry 6RGP without hydrogen atoms) of the protein immersed into an electrolyte solution. The energy difference at the proton transfer was described as a difference of electrostatic energies for different positions of the hydroxonium ion, which was described as a spherical cation at positions of crystallization water inside the protein molecule or arbitrary located far away from the macromolecule. The PB finite-difference grid spacing was 0.75 Å. The 0.12 M monovalent electrolyte solution was used. Its dielectric constant was 78.5. The thickness of ion exclusion layer around the macromolecule atoms (and the hydroxonium cation) was 2.5 Å. The dielectric interface with solution was shifted 0.8 Å away from the VdW radius of atoms. The VdW radius of the hydroxonium cation was set to 1 Å, and also to 1.5 Å for comparison. The partial charges of the protein atoms were described as only charged groups, and also as partial charges from the Amber ff99SB force field [34]. The PB calculations were done at a protein dielectric constant of 2, 4, and 10, as its true value is unknown (the value of 4 is regarded as the most plausible one). All PB calculations were carried out using the software developed by one of the authors of this article [35].

## 5. Conclusions

The reported results are in accordance with the scheme of the molecular oxygen reduction discussed [6]. That scheme was entirely deduced from interpretation of serial X-ray data. In the present work, we have confirmed/corrected what is suggested in the scheme protonation of oxygen ligands W1, W2, and W3 in the NI and reduced states of the TNC determined from X-ray data.

One definite correction is a hydroxyl ion instead of a water molecule at position W3 in the oxidized NI state. It just means that this ion will be protonated later when released from the T3 channel at the reduction of the TNC copper ions. It is shown that the central oxygen ligand W1 becomes a hydroxyl ion in the TNC oxidized state if oxygen ligand at position W3 has become a water molecule. However, the lack of accuracy for the estimation of W3 ligand protonation does not allow us to draw a definite conclusion (oxygen ligand W1 may remain ion O^2−^ in the NI state). What we can conclude is that after cleavage of the peroxide bond, one migration of a proton to oxygen ligand W3 takes place as was suggested in the scheme of the reaction [6]. One may admit the second migration of a proton in the fully oxidized state of TNC. The third proton migration described in the published scheme [6] is impossible.

Oxygen ligand W2 plays crucial role in the previously [5] and recently [6] suggested schemes of molecular oxygen reduction, but in the two schemes, this role is quite different in respect to the peroxide bond cleavage. It has been confirmed that this ligand is a water molecule in the fully reduced state of the TNC. For the oxidized TNC, the reported calculations do not allow us to determine the W2 protonation precisely. At least these results do not reject its form as a water molecule. Even admitting a loss of a proton by this molecule when the TNC copper ions get oxidized, the suggested reaction scheme will not be changed. The proton can just migrate along the T2 channel. When the copper ions get reduced, migration in the opposite direction must take place and the W2 ligand becomes a water molecule.

Anyhow, the intermediate and short-living PI state and the cleavage of the peroxide covalent bond require further QM studies.

## Figures and Tables

**Figure 1 ijms-24-02990-f001:**
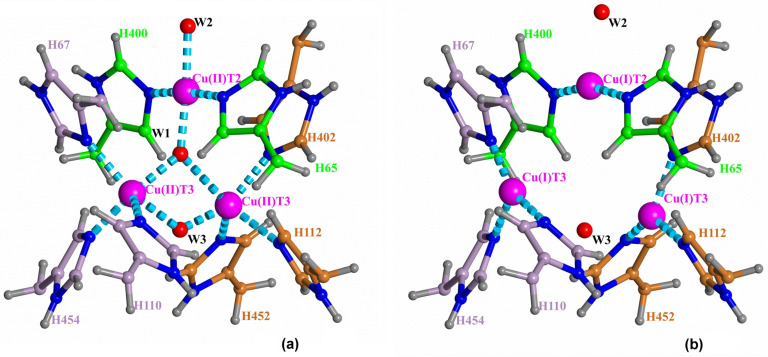
The oxidized (**a**) and reduced (**b**) states of the TNC [6,14]. Positions of copper ions (purple) are denoted as Cu(I)T3 and Cu(II)T3 for the T3 ions and Cu(I)T2/Cu(II)T2 for ion T2 for the reduced and oxidized states, respectively. There are also shown oxygen ligands (red spheres) and surroundings.

**Figure 2 ijms-24-02990-f002:**
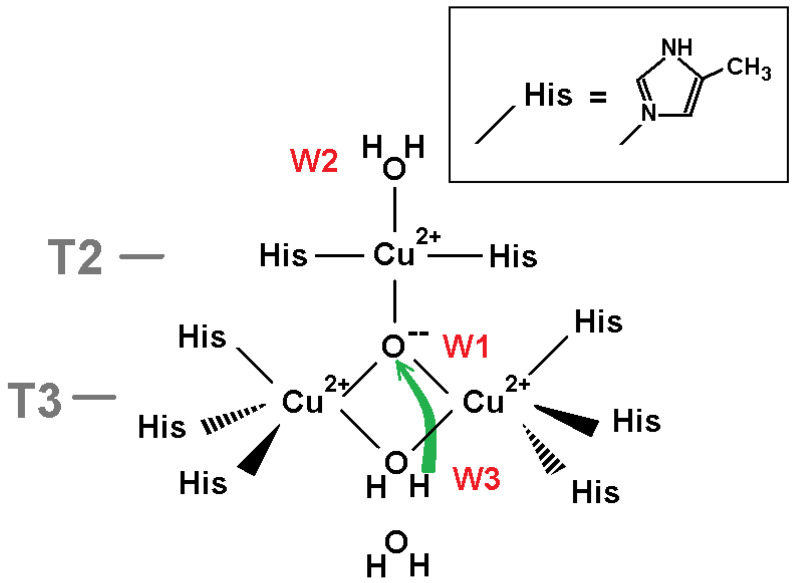
W1 protonation study model. The surroundings of the oxygen ligand W1 in the centre of the copper ions Cu(II) cluster that were chosen for QM calculations. The green arrow shows the proton transfer in the reaction H_2_O + O^2−^ → OH^−^ + OH^−^. Coordinating bonds are depicted as black lines. One more oxygen ligand (H_2_O) in channel T3 from the structure (pdb entry 6RGP) was included in the model.

**Figure 3 ijms-24-02990-f003:**
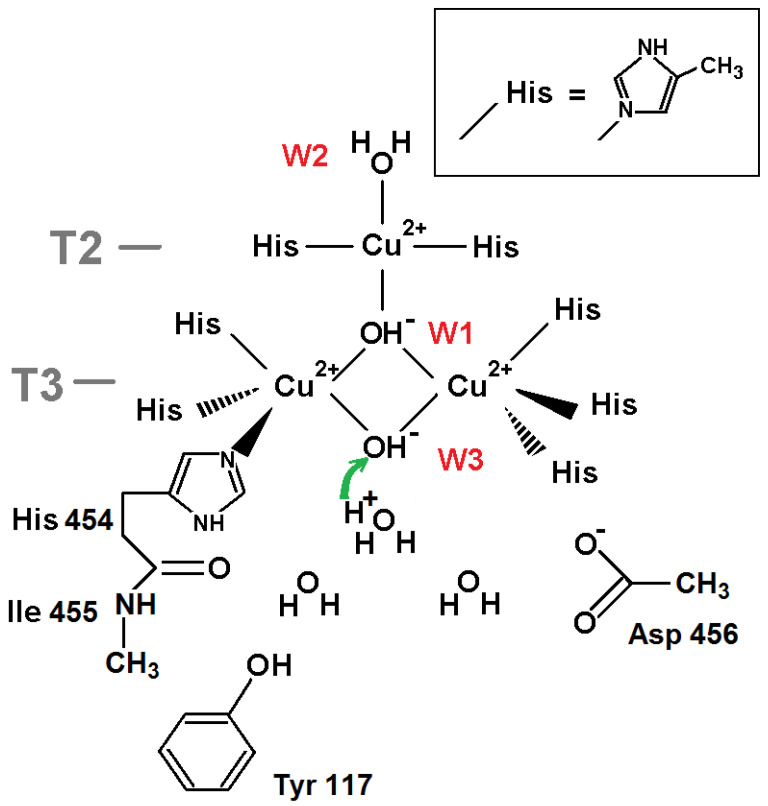
The TNC surroundings for DFT calculations of the proton transfer (green arrow) from hydroxonium cation H_3_O^+^ to the neighboring oxygen ligand W3 (OH^−^): H_3_O^+^ + OH^−^ → 2H_2_O. The central oxygen ligand is presented as a hydroxyl ion (OH^−^). The copper ion coordinating bonds and covalent bonds are depicted as black lines. Three more oxygen ligands in channel T3, in addition to the W3 ligand (pdb entry 6RGP), are added to model.

**Figure 4 ijms-24-02990-f004:**
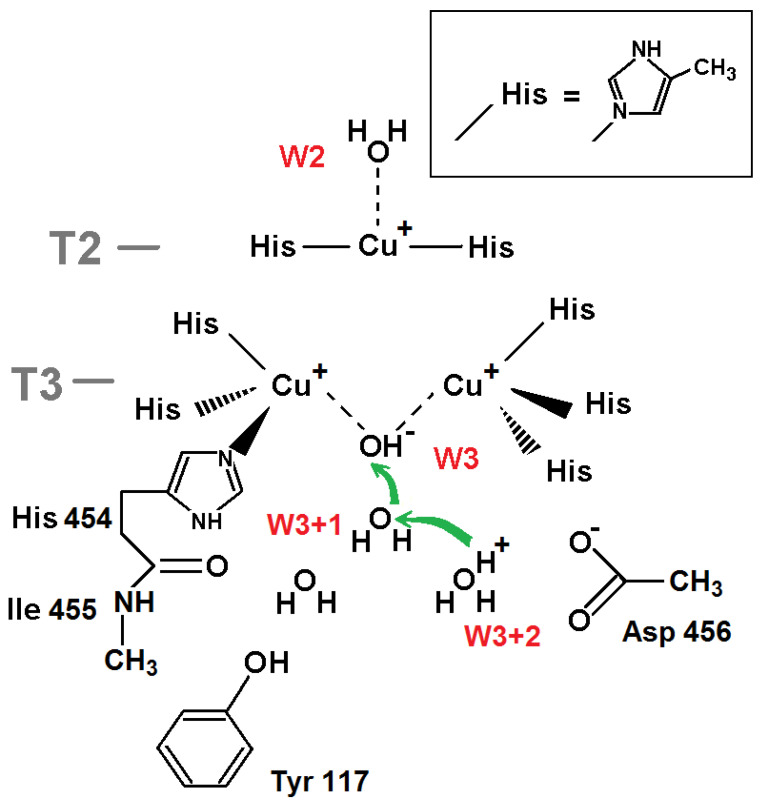
T3 channel environment chosen for DFT calculations of three states to explore protonation of oxygen ligand W3 when the TNC is reduced. The green arrows show two consequent transfers of a proton. The electrostatic interactions are shown as dotted lines. The coordinating and covalent bonds are shown as solid lines.

**Figure 5 ijms-24-02990-f005:**
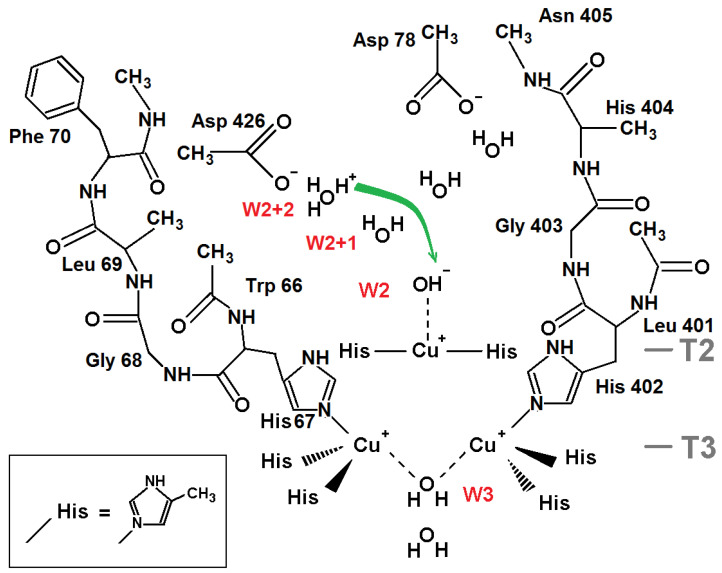
DFT calculations model chosen for a proton transfer (green arrow) along the water molecules chain of the T2 channel to OH^−^ at position W2(I) in the reduced state of TNC. Oxygen ligands H_3_O^+^, H_2_O, and OH^−^ (W2) correspond to the proton transfer chain, other two oxygen ligands in channel T2 are water molecules (all positions from pdb entry 6RGP). Purely electrostatic interactions are depicted as dashed lines. Coordinating and covalent bonds are shown as solid lines. See text for further details.

**Figure 6 ijms-24-02990-f006:**
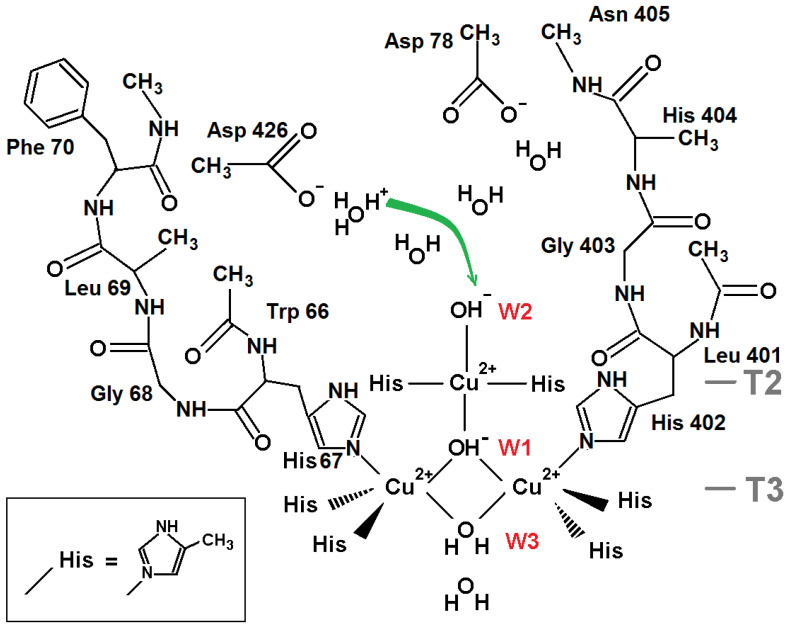
DFT calculations model chosen for a proton transfer along the water molecules chain of the T2 channel to OH^−^ at position W2(II) in the oxidized state of TNC. Oxygen ligands H_3_O^+^, H_2_O, and OH^−^ (W2) correspond to the proton transfer chain (marked by the green arrow), other two oxygen ligands are water molecules (all positions from pdb entry 6RGP). Coordinating and covalent bonds are shown as solid lines. See text for further details.

**Figure 7 ijms-24-02990-f007:**
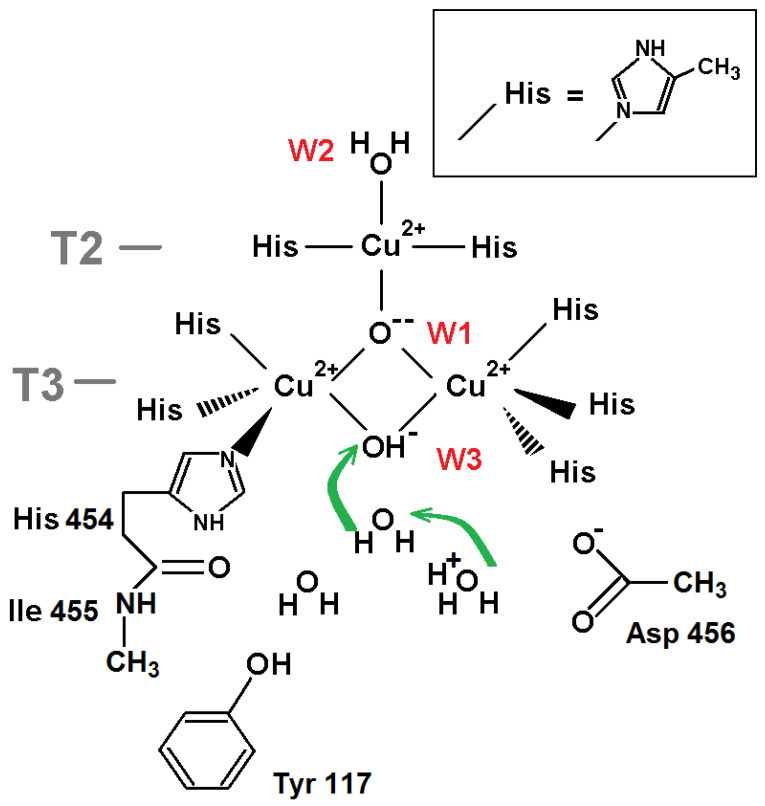
T3 channel environment chosen for DFT calculations of three states to explore protonation of oxygen ligand W3 when the TNC is oxidized and there is ion O^2−^ at position W1. The proton transfers are shown as green arrows. The copper ion coordinating bonds and covalent bonds are depicted as black lines.

**Table 1 ijms-24-02990-t001:** Energy differences Δ*E_ij_* = *E_j_* − *E_i_* (kcal/mol) between states *i* and *j* with different protonation of W1 and W3 ligand.

State *i*	State *j*	Δ*E_ij_*
(O^2−^,H_2_O)	transition state	18.4
transition state	(OH^−^up,OH^−^)	−45.5
(O^2−^,H_2_O)	(OH^−^up,OH^−^)	−27.1
(O^2−^,H_2_O)	(OH^−^down,OH^−^)	−21.6
(OH^−^up,OH^−^)	(OH^−^down,OH^−^)	5.5

**Table 2 ijms-24-02990-t002:** Energy differences Δ*E_ij_* = *E_j_* − *E_i_* (kcal/mol) between states *j* and *i* with different protonation of oxygen W3 and neighboring oxygen ligands shown in Figure 4.

State *i*	State *j*	Δ*E_ij_*
(H_3_O^+^,H_2_O,OH^−^)	(H_2_O,H_3_O^+^,OH^−^)	−13.0
(H_2_O,H_3_O^+^,OH^−^)	(H_2_O,H_2_O,H_2_O)	−18.2
(H_3_O^+^,H_2_O,OH^−^)	(H_2_O,H_2_O,H_2_O)	−31.2

**Table 3 ijms-24-02990-t003:** Energy differences Δ*E_ij_* = *E_j_* − *E_i_* (kcal/mol) between states *j* and *i* with different protonation of oxygen W2 and neighboring oxygen ligands shown in Figure 5.

State *i*	State *j*	Δ*E_ij_*
(H_3_O^+^,H_2_O,OH^−^)	(H_2_O,H_3_O^+^,OH^−^)	5.3
(H_2_O,H_3_O^+^,OH^−^)	(H_2_O,H_2_O,H_2_O)	−41.5
(H_3_O^+^,H_2_O,OH^−^)	(H_2_O,H_2_O,H_2_O)	−36.2

**Table 4 ijms-24-02990-t004:** Energy differences Δ*E_ij_* = *E_j_* − *E_i_* (kcal/mol) between states *j* and *i* with different protonation of oxygen W2 and neighboring oxygen ligands shown in Figure 6.

State *i*	State *j*	Δ*E_ij_*
(H_3_O^+^,H_2_O,OH^−^)	(H_2_O,H_3_O^+^,OH^−^)	22.9
(H_2_O,H_3_O^+^,OH^−^)	(H_2_O,H_2_O,H_2_O)	−19.1
(H_3_O^+^,H_2_O,OH^−^)	(H_2_O,H_2_O,H_2_O)	3.8

**Table 5 ijms-24-02990-t005:** Energy differences Δ*E_ij_* = *E_j_* − *E_i_* (kcal/mol) between states *j* and *i* with different protonation of oxygen W3 and neighboring oxygen ligands shown in Figure 7.

State *i*	State *j*	Δ*E_ij_*
(H_3_O^+^,H_2_O,OH^−^)	(H_2_O,H_3_O^+^,OH^−^)	−7.1
(H_2_O,H_3_O^+^,OH^−^)	(H_2_O,H_2_O,H_2_O)	−2.9
(H_3_O^+^,H_2_O,OH^−^)	(H_2_O,H_2_O,H_2_O)	−10.0

## Data Availability

Data sharing not applicable.

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
