# Peer review of "Quantum Mechanical Study of Oxygen Ligands Protonation for the Stable States of the Laccase Active Site"

_ijms, 2023, doi:10.3390/ijms24032990_

Round 1
Reviewer 1 Report
Present paper is nice example of multidisciplinary approach. Applying methods of computational and physical chemistry authors solved the problems arising in the field of molecular biology.
Introduction part is quite comprehensive and contains all necessary information about structure and properties of laccase oxidase. Results and discussion parts are devoted to the investigation of different protonation sites and mechanisms. This investigation was performed on a good theoretical level.
Minor comments/questions/recommendations:
Small paragraph of applicability of DFT-calculations to molecular biology issues should make present work more interesting for readers.
Overall, this paper is an example of good research. However, after reading the work repeatedly, there is a sense of a lack of global purpose. It looks like a pedagogical standard in the field of theoretical chemistry. What is the global goal of this study? What is purpose of the research of proton migration in trinuclear Cu ions? I convince that adding of extra motivation to introduction and especially conclusion part will make this work is fruitful for all readers.
Author Response
In reply to reviewer’s note that “Small paragraph of applicability of DFT-calculations to molecular biology issues should make present work more interesting for readers”, a piece of text with new references has been added at the beginning of section 4 “Results of calculations for the continuum-electrostatics model”, where applicability of DFT to proteins and its limitations are mentioned. Now it goes: “Despite wide applications of DFT… to redox processes [ ], evaluations…” (page 13).
The referee writes: “after reading the work repeatedly, there is a sense of a lack of global purpose. … What is the global goal of this study? What is purpose of the research of proton migration in trinuclear Cu ions? I convince that adding of extra motivation to introduction and especially conclusion part will make this work is fruitful for all readers.”
In reply to this concern, a phrase was added to the Introduction section and the short section Conclusions has been rewritten. Now at the very end of section Introduction there is one more sentence to clarify the objectives of the study: “Based on determined protonation of the TNC oxygen ligands, the scheme of the oxygen reduction mechanism has been improved” (page 4, top).
Now section Conclusions not merely briefly summarizes results of the study but also emphasizes how they relate to the scheme of molecular oxygen reduction in laccases published previously.
Reviewer 2 Report
The manuscript “Quantum Mechanical Study of Oxygen Ligands Protonation for the Stable States of the Laccase Active Site” by Gavryushov et al. reports DFT calculations on a moreld sytem of the laccase active site. The focus of interest are the protonation states and associated proton transfers between ligand of the coper ions in this active site.
Though not being able to conclude on the protonaion state of one ligand, the present work provides some insight into the protonation pattern of the laccase active site in reduced and oxidies form.
The paper is well written and can be recommended for publication subject to the following points have been addressed:
According to the methods section, only the positions hydrogen atoms, oxygen ligands, and capping groups were optimised (line 168). While this makes sure, the optimised model is as close to the crystal structure as possible it deprives the authors of the opportunity to also use the optimised geometries for evaluating the likelihood of a certain protonation state. In the protein environment, the active site is flexible and could rearrange as a response to a change in protonation state. This should at least be discussed. Relatedly, do the authors observe changes in the positions of the oxygen ligands upon proton transfer?
It is not fully clear what is meant by "(W3+2,W3+1,W3) with W3+1 and W3+2 denoting first and second oxygen ligand from the W3 oxygen ligand in the channel" (line 291). And similarly in line 300. There is only one W3 lor W2 igand, respctively. Do the authors mean proton positions at the respective oxygen atoms? Or are other ligands considered? Which?
Author Response
The reviewer writes: “... the optimised model is as close to the crystal structure as possible it deprives the authors of the opportunity to also use the optimised geometries for evaluating the likelihood of a certain protonation state. In the protein environment, the active site is flexible and could rearrange as a response to a change in protonation state. This should at least be discussed.”
In reply to this referee’s concern, it should be noted that we used two structures interpreted from X-ray data at subatomic resolution. One is for the oxidized state of TNC and the second one is for its fully reduced state. Both are clearly distinguished for positions of the copper ions and oxygen ligands in the TNC, whereas the positions of the side chains of coordinated histidines remain intact. This difference is shown in Fig. 1 completely drawn from X-ray data (pdb code 6RGP). According to the methodology chosen, the evaluation of the ligand’s protonation state follows from change of the system energy at a change of ligand protonation while copper ions and other atoms are correctly placed according to X-ray data in the state considered. The change of energy follows from the proton transfer between two neighboring positions of oxygen ligands (about 3 A). The copper ions position errors in the X-ray data are much lower (~0.04 A). The ligand position is subjected to optimization and all model errors are attributed to this position (it was not deviated more than ~0.1 A from X-ray data except for one case mentioned below). In cases when the proton position could affect the system energy, a particular study is done (section 3.1).
In reply to the reviewer’s question: “do the authors observe changes in the positions of the oxygen ligands upon proton transfer?”, it should be said that such analysis was done. Only in one case (for ligand W2 as a hydroxyl ion in the reduced state of TNC) there was detected a significant deviation from X-ray data suggesting in favor of water molecule there (p. 3.4, page 10).
In reply to the reviewer’s note that "it is not fully clear what is meant by <<(W3+2,W3+1,W3) with W3+1 and W3+2 denoting first and second oxygen ligand from the W3 oxygen ligand in the channel>>… and similarly in line 300… Do the authors mean proton positions at the respective oxygen atoms? Or are other ligands considered?", Figures 4 and 5 have been corrected. Now they include labels of oxygen ligands W3+1 and W3+2, as well as W2+1 and W2+2 to make the presentation clearer.